# ADVERSARIAL INSTANCE ATTACKS FOR INTERACTIONS BETWEEN HUMAN AND OBJECT

## ABSTRACT

Adversarial attacks can easily deceive deep neural networks (DNNs); however, they are also valuable for evaluating the robustness of DNNs. Existing attacks on object detection primarily focus on attacking the recognition of objects, while whether the attacks remain effective on more complex scene understanding tasks (*e.g.*, extracting the interaction between objects) remains largely unexplored. In this paper, we, for the first time, propose *Adversarial Instance Attacks*, a novel black-box attacking framework for scene interactions without interfering with object detections. To achieve the goal, we first introduce an Interaction Area Sampling module that identifies vulnerable anchors (area) for adversarial instances positioning. Secondly, we design an Object Category Search module and build an interaction co-occurrence knowledge graph to explore categories with higher obfuscation scores toward specific object-interaction pairs. Finally, our framework generates perturbations that serve as adversarial instances with high co-occurrence obfuscation towards specific interactions in vulnerable areas and deceive HOI models. Extensive experiments conducted against multiple models demonstrate effectiveness in attacking interactions of HOI. Our approach surpasses existing methods by significant margins, achieving an improvement of at least *+10.36%*.

## 1 INTRODUCTION

Deep learning has made significant contributions to a variety of vision and language tasks, including detection (Lin et al., 2017), classification (Wang et al., 2021b), scene understanding (Xu et al., 2017), *etc*. However, DNNs are highly vulnerable to adversarial attacks (Goodfellow et al., 2015; Szegedy et al., 2014). These visually imperceptible perturbations could deceive DNNs into wrong predictions, which poses significant challenges to both digital and physical world applications (Liu et al., 2023; 2019; 2020a; Wang et al., 2021a).

By contrast, adversarial examples are also beneficial for evaluating and better understanding the robustness of DNNs (Zhang et al., 2020; Xie et al., 2020; Liu et al., 2022; Tang et al., 2021). In the past years, extensive research has been devoted to attacking object detection where adversaries aim to attack the recognition of objects; however, the robustness of more complex scene understanding tasks (*e.g.*, inferring the interaction between contents in the scene (Yang et al., 2023; Tong et al., 2022)) remains under-explored. Interaction-based scene understanding aims to capture spatial or semantic relationships corresponding to contents, providing a linking bridge between low-level scene perception, such as detention (Zhang et al., 2022a) and classification (Deng et al., 2009), and high-level scene understanding, such as visual question answering (Lee et al., 2019) and scene captioning (Yan et al., 2022; Shi et al., 2022).

However, it is highly non-trivial to simply extend current attacks on object detection to interactions. Object detection attacks (Szegedy et al., 2014) aim to deceive detectors into making wrong predictions on objects by mainly generating multiple bounding boxes for additional object predictions (Chen et al., 2017; Brendel et al., 2017; Maho et al., 2021); however, the relationships between these generated object predictions are comparatively low, which may fail to influence the interaction prediction process. In addition, these attacks are more easily perceived by object detection defenses, which in turn reduce the attacking abilities.

In this paper, we take the first step towards adversarial attacks on the interaction between scene contents and realize based on the Human Object Interaction (HOI) task (Gkioxari et al., 2018). To

address the problem, we propose a novel attacking format dubbed *Adversarial Instance Attacks (AIA)*, which deviates interactions with vulnerable information. To achieve the goals, our AIA is constructed by the Interaction Area Sampling (IAS) module to discover vulnerable object anchors (areas), the Object Category Search (OCS) module to assign the most confusing object category to the area via prior-guided searching, and the Adversarial Instance Generation (AIG) module to determine attacking effects of the combined candidate areas and object categories. We formulate the successful interaction attack protocol as generating fake instances specifically for interactions without misleading humans and objects. Therefore, our generated perturbations could serve as an adversarial instance with higher interaction superiority over the original ones and deceive models to make wrong predictions. Extensive experiments against multiple models demonstrate that our proposed framework successfully attacks interactions and outperforms others significantly.

The main **contributions** of this paper are summarized as:

- This paper marks the initial stride toward adversarial attacks on scene interactions, setting it apart from previous adversarial attacks.

- The proposed black-box Adversarial Instance Attacks framework delves into both spatial and semantic aspects, searching for potentially fragile information to construct adversarial instances for attacks.

- AIA framework effectively compromises adversarial attacks on interactions and demonstrates its efficacy that is supported by extensive experiments conducted on multiple models (at least *+10.36%*).

## 2 RELATED WORK

**Adversarial Attack on Object Detection.** There has been a lot of research on object detection attacks. White-box attacks can obtain the overall parameters of the model during perturbation attacks. Szegedy et al. (2014) proposed the concept of gradient fragility in neural networks. Subsequently, classic white-box attack methods such as BIM (Kurakin et al., 2017), PGD(Madry et al., 2018), FGSM (Goodfellow et al., 2015), I-FGSM (Kurakin et al., 2017) and C&W (Carlini & Wagner, 2017) were introduced. In score-based attacks, Techniques such as ZOO (Chen et al., 2017), SimBA (Guo et al., 2019), N-Attack (Li et al., 2019a). In decision-based attacks, the field has seen steady progress since the advent of Boundary Attack (Brendel et al., 2017). Significant works have successively emerged, including HSJA (Chen et al., 2020), GeoDA (Rahmati et al., 2020), QEBA (Li et al., 2020a) and Surfree (Maho et al., 2021). Moreover, Transfer-based attack (Papernot et al., 2016; Liu et al., 2016; Wang & He, 2021; Yuan et al., 2021), Substitute-based attack (Zhou et al., 2020; Wang et al., 2021c; Li et al., 2020b), and pixel attack (Su et al., 2019) have all demonstrated notable attack performance.

There have been discussions on various aspects beyond the foundational task of object detection attacks. Liu et al. (2017) enhanced the attack transferability . UAP (Moosavi-Dezfooli et al., 2017) compute a universal perturbation. Zhang et al. (2022b) employing dual attention-based targeted feature space attack. However, there is limited research specifically focused on interaction attacks. Additionally, traditional object detection attacks may not be suitable for this task (as indicated in the experiments conducted in section 4.2). Cai et al. (2022) enhances the success probability by influencing the categories of other objects in the one image. This commendable research takes into account the relationships between instances. The difference in our approach lies in the fact that the attack target is the relationship itself.

**Interactions between Human and Objects.** HOI task aims to detect the interactions between humans and objects in an image (Gkioxari et al., 2018). Prior information from external sources has been shown to be helpful in distinguishing interactions between humans and objects (Lu et al., 2016). Kato1 et al. (2018) utilized external knowledge graphs and CNN to learn how to combine verbs and nouns for interactions. Kim et al. (2020) address the long-tail problem in HOI dataset by modeling the natural correlation and anti-correlation of human-object interactions. Li et al. (2019b; 2022a) bridging the differences between datasets and making the model more generalizable. Liu et al. (2020b) address the issue of imbalanced samples during training. Yuan et al. (2022a;b) proposes that objects play an important role in capturing interaction information and can effectively guide

the model to recognize interactions. There are also some works that focus on the interaction area. Gao et al. (2018) uses an instance-centric attention module to dynamically highlight interaction area in the image. Wang et al. (2020) proposes using CNN to directly detect the interaction with human-object pairs. PPDM[21] (Liao et al., 2020) posits that the interaction point is the midpoint of the line connecting the midpoint of the human and the object box. DIRV (Fang et al., 2021) focuses more on the densely interactive region within each human-object pair. GGNet(Zhong et al., 2021) mimics the human process of glance and gaze. QAHOI (Chen & Yanai, 2021) extracts features at different spatial scales through a multi-scale architecture. FGAHOI (Ma et al., 2023) extracts features of human, object, and interaction areas from different sub-levels of spatial information.

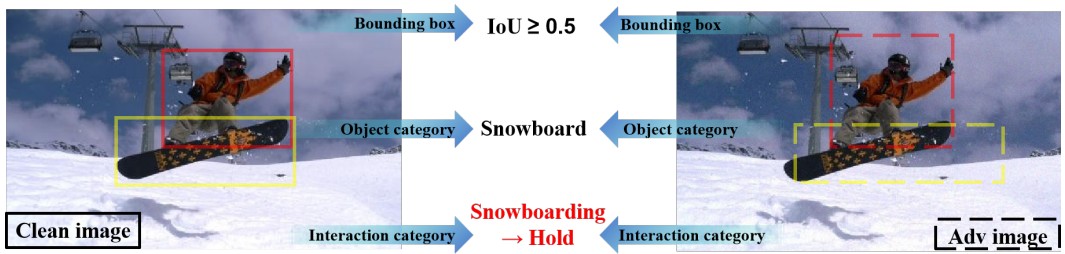

Figure 1: Illustration of adversarial attacks for interactions, which only change the predicted interaction (from "snowboarding" to "hold") while remaining0 the object detection.

## 3 METHODOLOGY

To interfere with interactions in the scene, we proceed from the view of the Human-Object Interaction (HOI) task, in which the interaction is represented as a triplet including the human and its location, the object location and its category, and the interaction category. We first provide the definition of the Adversarial Interaction Attack problem, which aims to disturb the interaction prediction of a pair of humans and objects without changing predicted object categories as shown in Figure 1. Under this problem set, the attack is difficult to capture by directly analyzing scene contents, thus, bringing in confusion for high-level scene understanding.

### 3.1 PROBLEM DEFINITION

**Human Object Interaction.** Given an input image $\mathbf{x} \in \mathbb{R}^{H \times W \times 3}$, the HOI model (Kim et al., 2021) predicates the output as $\{\mathbf{h}_i^b, \mathbf{o}_i^b, \mathbf{o}_i^l, \mathbf{a}_i\}$, where $i$ is the indication of triplets. $\mathbf{h}_i^b \in \{0, 1\}^{1 \times 4}$ represents the bounding box of the human, $\mathbf{o}_i^b \in \{0, 1\}^{1 \times 4}$ represents the bounding box of the object instance. $\mathbf{o}_i^l \in \{0, N^o - 1\}^{1 \times N^o}$ represents the categories of the object, where $N^o$ is the number of object categories of the dataset (the V-COCO dataset(Gupta & Malik, 2015; Lin et al., 2014)). $\mathbf{a}_i \in \{0, N^a - 1\}^{1 \times N^a}$ represents the categories of the object, where $N^a$ is the number of interaction categories in the dataset. The loss function of HOI can be formulated as:

$$\mathcal{H}_{\text{match}}(\rho) = \mathcal{H}_b(\rho) + \mathcal{H}_u(\rho) + \mathcal{H}_c(\rho) + \mathcal{H}_a(\rho) = \sum_{i=1}^{N^t} h_{\text{match}}(y_i, \hat{y}_{\rho(i)}). \quad (1)$$

$\mathcal{H}_{match}$ calculates the matching cost and consists of four types of costs: the bounding box regression cost $\mathcal{H}_b$, the IoU cost $\mathcal{H}_u$, the object category cost $\mathcal{H}_c$, and the interaction category cost $\mathcal{H}_a$. $h_{\text{match}}(y_i, \hat{y}_{\rho(i)})$ calculates cost between triplet groundtruth $y_i$ and a prediction with indication of $\rho(i)$. $N^t$ is the number of triplets in the image. The optimization is conducted by the Hungarian algorithm, as follows:

$$\hat{\rho} = \arg \min_{\rho \in \Theta} (\mathcal{H}_b(\rho) + \mathcal{H}_u(\rho) + \mathcal{H}_c(\rho) + \mathcal{H}_a(\rho)), \quad (2)$$

where $\Theta$ is a search space for $\rho$.

**Attacks on HOI.** To conduct the adversarial interaction attack, our goal is to bring confusion on the interaction of a pair of human and an object, without contaminating their detection results including

the human bounding box, object bounding box, and object category. Based on this consideration, the optimization of the adversarial interaction attack is defined as:

$$
\begin{aligned}
\hat{\phi} &= \arg \max_{\phi \in \Theta_{N^a}} \mathcal{H}_a(\phi|\hat{\rho}), \\
s.t.\ \hat{\rho} &= \arg \min_{\rho \in \Theta_{N^o}} (\mathcal{H}_b(\rho) + \mathcal{H}_u(\rho) + \mathcal{H}_c(\rho)),
\end{aligned}
\tag{3}
$$

where $\Theta_{N^a}$ is the search space for a permutation of $N^a$ interactions, so as $\Theta_{N^o}$.

In contrast to identifying objects in the scene, identifying interactions has no direct response area, therefore bringing more challenges during perception. Given the target model triplet prediction $\{\mathbf{h}_i^b, \mathbf{o}_i^b, \mathbf{o}_i^l, \mathbf{a}_i\}$, a direct strategy to disturb the interaction is adding noises on the union area of human and object, following the feature extraction procedure in HOI. However, directly adding noise to the union area not only drop the precise of interaction perception but also bring interference in locating and classifying human and objects. It also deviates from the motivation of the adversarial interaction attack. Previous research for HOI (Ma et al., 2023; Wu et al., 2023) explores interactive affinity for interactions, however depending on an ill-posed assumption(position center (Liao et al., 2020)) or additional perception (human pose or parsing (Liu et al., 2020b; Li et al., 2019b)).

## 3.2 OVERALL FRAMEWORK

Since it is difficult to conduct an interaction attack with its corresponding specific information, a straightforward thought is that we can add some manipulations so as to distract the attention of current interactions. Based on this, we design the attack pipeline as adding an invisible object to the image, whose interaction has superiority over the original object, so as to disturb the original interaction. The core elements are where to insert the invisible object and what is the category of the object.

To achieve the goal, we formulate the framework as a black-box attacking procedure and put forward the Adversarial Instance Attack (AIA) based on instances holding vulnerable areas and categories to interactions. The AIA is constructed by the Interaction Area Sampling (IAS) module to discover vulnerable anchors (areas), the Object Category Search (OCS) module to assign the most confusing object candidate categories to the area via interaction-object pairs prior information of interactions (Lu et al., 2016), and the Adversarial Instance Generation (AIG) module to determinate attacking effects of the generated adversarial instances, i.e., the combined candidate areas and object categories. The AIA pipeline is illustrated in Figure 2. Given the clean image $x$, triplet predictions $\mathcal{P}$, including human location, object location and category, and interaction category, are obtained through a target HOI model(Kim et al., 2021; Liao et al., 2022). Then the clean image $x$ and the triplet with top-1 confidence are fed into the IAS module to generate $N^q$ candidate interaction areas via a multi-scale sampling strategy. To obtain categories of candidate areas, the interaction $a$ of the triplet is sent into the OCS module to find target categories based on the constructed Interaction-Object Co-occurrence Graph (IOCG). Subsequently, we incorporate candidate areas and target categories to formulate adversarial instances and use the AIG module to determine if the instance attack makes effects.

## 3.3 INTERACTION AREA SAMPLING

Inspired by the anchor-based object detection transformer pipeline, we convert the adversarial object localization to provide meaningful anchor boxes in the image. Moreover, considering that objects in the HOI dataset always hold diverse sizes of areas (Yuan et al., 2022a), we utilize the multi-scale feature encoding structure to ensure the capture representations of objects with different sizes. Furthermore, we incorporate the global contents as an overall controller to provide global guidance to the anchor extraction therefore facilitating the accuracy.

During the image context encoding, the clean image is sent to the hierarchical backbone network $\Phi(\cdot)$ to obtain the multi-scale feature $\mathbf{z}_l \in \mathbb{R}^{\frac{H}{4 \times 2^l} \times \frac{W}{4 \times 2^l} \times 2^l c_s}$, where $l$ is the indication of layers, $c_s$ is the channel number of features. $\mathbf{z}_l, \forall l \in [1, L]$ are then merged via $1 \times 1$ convolution, flatten, and concatenation operations, providing feature representation $\mathbf{M}$. During the anchor decoding procedure, the positional embedding $\mathbf{P}$ is incorporated with $\mathbf{M}$ and global content embedding $\mathbf{C}$, separately, providing updated representations. Then a cross-attention operation is conducted between updated multi-scale features and global content embedding, following a pre-trained multiple-layer perception

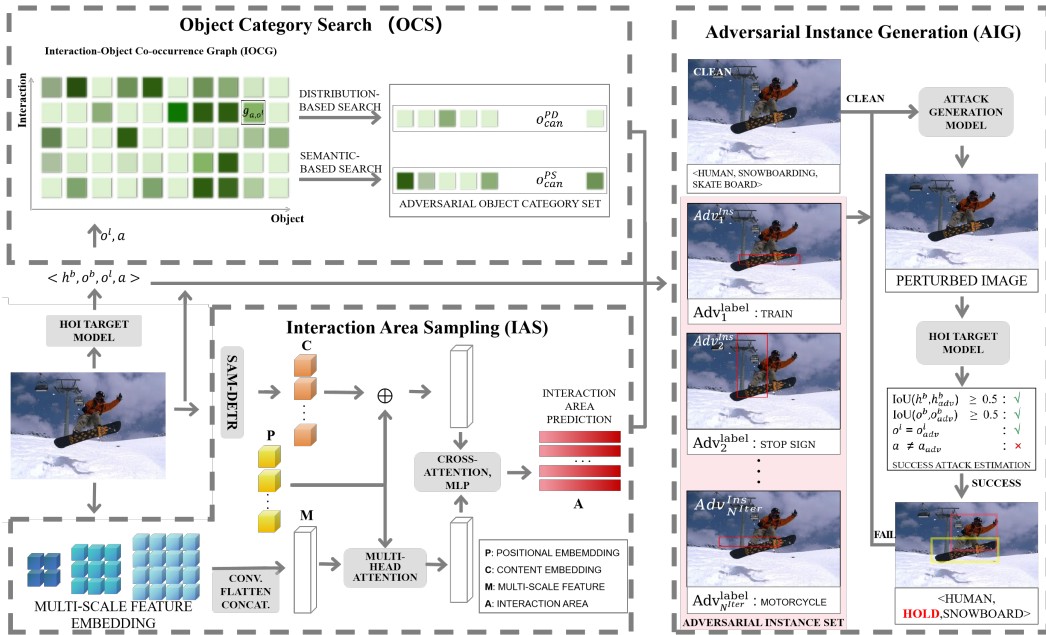

Figure 2: The overview of the proposed Adversarial Instance Attack (AIA). Given the clean input image, the AIA generates successful adversarial attacks for scene interactions via Interaction Area Sampling (IAS), Object Category Search (OCS), and Adversarial Instance Generation (AIG).

to provide the anchor boxes of interaction areas, i.e., $\mathbf{A} \in \mathbb{R}^{N^{t^*} \times 4}$, where $N^{t^*}$ is the number of predicted triplets.

## 3.4 OBJECT CATEGORY SEARCH

Given candidate interaction areas that contribute most to the interaction prediction, a strong attack is formulated as an instance located at the candidate area and holding the most confusing object category. Based on this consideration, the Object Category Search (OCS) module aims to find the most vulnerable object categories for candidates, so as to prevent the original interaction from being perceived. To achieve this goal, we follow the utilization of interaction-object co-occurrence(Li et al., 2019b; 2022a) and construct a complete co-occurrence matrix to represent the confidence of a pair of interactions and objects. Then we put forward two search strategies to choose object categories for candidate interaction areas.

**Interaction-Object Co-occurrence Graph Generation.** We collect data from three HOI datasets including HICO-DET(Chao et al., 2018), V-COCO, HAKE(Li et al., 2023; Lu et al., 2018), and PaStaNet(Li et al., 2020c). The dataset used for the attack is V-COCO, which consists of 80 object categories and 25 interaction categories. When aligning the content of other datasets with V-COCO, we used the intersection of categories. Then we count the number of co-occurrences between each pair of interactions and objects and normalize counting numbers into the range of $(0, 1)$. All counting numbers are stored in a matrix $\mathbf{G} = \{g_{n,m}\}^{N^a \times N^o}$, $g_{n,m} \in (0, 1)$ named Interaction-Object Co-occurrence Graph (IOCG), which serves as category prior information.

**Category Search Based on Distribution Prior.** Based on the long-tailed predicate distribution statistics of HOI datasets(Li et al., 2022b; Wan et al., 2019; Lee et al., 2018), interactions presented in the tailed area are associated with few training samples. Under this circumstance, representations of interaction-object combinations in the tail are difficult to learn, therefore bringing vulnerability to the HOI model. Following this observation, we put forward a category-search strategy based on the view of category distribution. Given the predicted interaction $a$ from the target model, the association of this interaction with $N^o$ object categories is represented as $g_{\mathbb{I}(a),m}, \forall m \in [1, N^o]$ and $\mathbb{I}(a)$ is the

indication of interaction $a$. With the interaction association, the searched object categories are defined as:

$$\mathbf{s}^D = \{m | g_{\mathbb{I}(a),m} < g_{\mathbb{I}(a),\mathbb{I}(o^l)}, m \in [1, N^o]\}. \tag{4}$$

**Category Search Based on Semantic Prior.**    The HOI task always presents label confusion among interaction categories, indicating that human and object pairs can be described by different interaction categories holding similar semantic information, such as "holding" and " playing" for <human, skating board>.  Under this consideration, we can find a substitute object category whose co-occurring interaction has the closest similarity with the interaction $a$ from the target model output. To find this substitute object category, we design a category-search strategy based on semantic information of interaction pairs by selecting a closely semantic-related interaction category. Given the object category $o^l$ predicted by the target model, its prior vector is defined as $\mathbf{g}_{*,\mathbb{I}(o^l)} \in \mathbb{R}^{1 \times N^a}$, which describes the relevance of $o^l$ to the $N^a$ interactions.

To obtain vulnerable object categories based on object categories of the clean image, we first extract the potentially vulnerable interaction categories whose co-occurring scores are lower than the clean pairs and store these categories in $\mathbf{s}^{ind}$. Then for each vulnerable interaction category, we pick out object categories whose co-occurring scores are also lower than the clear pairs and store them in the category indication vector $\mathbf{s}^S$. The procedure is illustrated as:

$$\mathbf{s}^S = \bigcup_{n \in \mathbf{s}^{ind}} \{m | \arg \min_m g_{n,m} < g_{\mathbb{I}(a),\mathbb{I}(o^l)}\},$$
$$\mathbf{s}^{ind} = \{n | g_{n,\mathbb{I}(o^l)} < g_{\mathbb{I}(a),\mathbb{I}(o^l)}, \ n \in [1, N^a]\}. \tag{5}$$

## 3.5    Adversarial Instance Generation

With the obtained candidate interaction areas and adversarial object categories that are vulnerable to target model prediction, we propose an Adversarial Instance Generation module to generate adversarial instances of potential adversarial attacks for interactions.

**Adversarial Instance Generation.**    Given the extracted candidate predicate areas $\mathbf{A}$. To ensure that the adversarial instances are high-effect to the top-1 triplet, we re-order the sequences of candidate areas in the set, based on the distance away from the triplet. To this end, a distance calculation function is defined as

$$d(c_1, c_*) = f_{\min}(f_{\mathrm{mid}}(h^b, o^b), \mathbf{A}_{i,*}).$$

$f_{\mathrm{mid}}(\cdot)$ calculates the midpoint between human bounding box $h^b$ and object bounding box $o^b$. $f_{\min}(\cdot)$ calculates the Euclidean distance between the midpoint and the center point of a candidate area. After that, all candidate areas in $\mathbf{A}$ are sorted in an ascending order based on $d(c_1, c_*)$ to obtain $\hat{\mathcal{A}}$. All candidates in $\hat{\mathcal{A}}$ are then assigned a vulnerable object category coming from the OCS module ($\mathcal{O}(\mathbf{s}^D)$ or $\mathcal{O}(\mathbf{s}^S)$). The adversarial instance set $\mathcal{I}_{\mathrm{Adv}}$ is constructed as $\mathcal{I}_{\mathrm{Adv}} = \{\mathrm{Adv}_1^{\mathrm{Ins}}, \cdots, \mathrm{Adv}_{N^{\mathrm{Iter}}}^{\mathrm{Ins}}\}$, where $\mathrm{Adv}_{N^{\mathrm{Iter}}}^{\mathrm{Ins}} = (\mathrm{Adv}_{N^{\mathrm{Iter}}}^{\mathrm{label}}, \mathrm{Adv}_{N^{\mathrm{Iter}}}^{\mathrm{bbox}})$. Noticing that the number of candidate areas is larger than that of confused object categories, therefore bringing a lack of object categories. To tackle this issue, we conduct a random assignment to link confused object categories to candidate predicate areas to generate the attack instance.

**Adversarial Attacks.**    Given the adversarial instance set $\mathcal{I}_{\mathrm{Adv}}$, each pair of the adversarial instance $\mathrm{Adv}_1^{\mathrm{Ins}}$ and the clean image $x$ is sent into an attack generation model (Targeted adversarial Objectness Gradient attacks (TOG) (Chow et al., 2020b;a) in the paper) to generate perturbed image $x^{\mathrm{Adv}}$. To ensure the generalization of the attack, during generation, we conduct white-box attacks (HOG) based on two models Faster R-CNN(Ren et al., 2015) and YOLOv3(Redmon & Farhadi, 2018), whose network architectures are totally different with the target HOI models. After generating attacks, the adversarial images are evaluated by the target model and predicted with $\{h_{\mathrm{Adv}}^b, o_{\mathrm{Adv}}^b, o_{\mathrm{Adv}}^l, a_{\mathrm{Adv}}\}$. The adversarial attack generation stops until there is a successful attack, which is defined as:

$$\mathcal{F}_{AT}(x_i, x_i^{\mathrm{Adv}}) = \begin{cases} 1, & \text{if} \quad a \neq a_{\mathrm{Adv}} \\ 0, & \text{if} \quad a = a_{\mathrm{Adv}} \end{cases} \tag{6}$$

$$s.t. \quad \mathrm{IoU}(h^b, h_{\mathrm{Adv}}^b) \geq 0.5, \ \mathrm{IoU}(o^b, o_{\mathrm{Adv}}^b) \geq 0.5, \text{ and } o^l = o_{\mathrm{Adv}}^l,$$

where $\mathcal{F}_{AT}$ is the adversarial attack generation function.

## 4 EXPERIMENTS

### 4.1 EXPERIMENTAL SETUP

**Dataset and Model.** We collect the first 500 images from the V-COCO (Gupta & Malik, 2015; Lin et al., 2014) test set to compose the interaction attack dataset, which consists of $N^o = 80$ object categories and $N^a = 25$ interaction categories. To perform black-box attacks, we select HOTR (Kim et al., 2021) and GEN-VLKT (Liao et al., 2022) HOI models as target models.

**Adapting the HOI to Black Box.** The output from the HOI model is in the form of scores. Taking HOTR as an example, HOTR outputs a three-dimensional matrix with dimensions $< 25, N^h, 80 >$, where $N^h$ is the number of human bounding boxes. Values in the matrix represent the scores assigned to each combination. Since our attack type is black-box, it requires the target model to provide qualitative recognition results instead of scores. Our approach takes the recognition result with the highest score as the qualitative output of the model.

**Implementation Details and Evaluation Metric.** The prior information is derived from HICO-DET (Chao et al., 2018), V-COCO, HAKE (Li et al., 2023; Lu et al., 2018), and PaStaNet (Li et al., 2020c). The visual feature extractor in the IAS model is Swin-Tiny (Liu et al., 2021). We set the maximum perturbation level to $L_\infty \leq 10$. Global content embedding feature extraction utilizes the SAM-DETR (Zhang et al., 2022a). The number of adversarial instances is set to 100. To ensure the integrity of the object and human bounding boxes before and after perturbation, it requires that IoU $\geq 0.5$. During the evaluation, the effectiveness is measured based on the attack success rate, where the definition of attack success is derived from equation 6.

**Comparison Methods.** We conduct comparisons with transfer-based black-box adversarial attacks (Wang et al., 2023; Zhang et al., 2023; Li et al., 2022a) via employing I-FGSM (Kurakin et al., 2017) and PGD Madry et al. (2018) on white-box surrogate models (Two-stage detectors Faster R-CNN (Ren et al., 2015), one-stage detectors YOLOv3 (Redmon & Farhadi, 2018) and transformer-based DETR (Carion et al., 2020)).

### 4.2 RESULTS AND DISCUSSION

**Experimental Result Analysis.** We present the success rates of various attack strategies in Table 1. To begin with, we evaluate the *performance of different attack methods*, namely I-FGSM (Kurakin et al., 2017) and PGD (Madry et al., 2018), in terms of their impact on target models. It is noteworthy that I-FGSM consistently exhibits a lower rate of successful attacks and maintains a stable trend across different target and surrogate models. For instance, when targeting the surrogate model YOLOv3, I-FGSM achieves relatively low success rates (0.63% and 25.43% for 1 iteration and 100 iterations, respectively), with marginal differences observed between the two HOI models (less than 8%). Considering the definition of a successful attack, we posit that I-FGSM's effectiveness is limited in interactions, primarily due to its pronounced disruption of objects. Furthermore, we compare *results across three surrogate models*. Notably, the two-stage surrogate model Faster R-CNN displays greater vulnerability to interaction-based attacks compared to YOLOv3 and DETR (e.g., 9.55% for Faster-RCNN as opposed to 0% for YOLOv3 and 0.99% for DETR in the OGD attack on the GEN-VLKT target model). This susceptibility can be attributed to the robustness of the two-stage object detection pipeline employed by Faster R-CNN. Moreover, we scrutinize the *attack outcomes on various target models* and draw the conclusion that GEN-VLKT is more susceptible to attacks than HOTR. This vulnerability can be attributed to GEN-VLKT's incorporation of features provided by CLIP (Radford et al., 2021), whereas HOTR does not leverage such features. Although CLIP enhances HOI recognition performance, it also renders the model more susceptible to the long-tail effect, consequently making it more prone to successful attacks when exposed to adversarial instances.

Compared with the previous attack methods on interactions, our proposed AIA achieves the expected effects on attacking interactions based on GEN GEN-VLKT and HOTR. When the adversarial attack iteration is 1, the attack for GEN-VLKT achieves 55.17%, and HOTR is at 18.67% and reaches 76.35% and 67.41% when the iteration is 100. The AIA achieves outstanding successful attack rate

Table 1: Attack success rate under perturbation $L_\infty \leq 10$.

| Target Model | Attack | Adversarial Attack Iteration | | |
|---|---|---|---|---|
| | | 1 | 50 | 100 |
| GEN-VLKT | I-FGSM (Faster R-CNN) | 0.99 | 23.65 | 30.54 |
| | I-FGSM (YOLOv3) | 0.98 | 23.53 | 31.37 |
| | PGD (Faster R-CNN) | 0.00 | 4.43 | 7.91 |
| | PGD (DETR) | 0.99 | 23.15 | 30.54 |
| | PGD (YOLOv3) | 0.00 | 5.42 | 9.36 |
| | PGD (YOLOv3 & Faster R-CNN) | 9.80 | 63.73 | 67.65 |
| | **Ours** | **55.17** | **76.35** | **76.35** |
| HOTR | I-FGSM (Faster R-CNN) | 0.63 | 17.72 | 25.32 |
| | I-FGSM (YOLOv3) | 0.63 | 17.72 | 23.42 |
| | PGD (Faster R-CNN) | 9.49 | 52.22 | 55.38 |
| | PGD (DETR) | 0.32 | 18.67 | 26.27 |
| | PGD (YOLOv3) | 8.23 | 55.70 | 61.08 |
| | PGD (YOLOv3 & Faster R-CNN) | 5.38 | 49.68 | 56.33 |
| | **Ours** | **18.67** | **59.49** | **67.41** |

increment over others. We conclude that the AIA is specially designed for interaction perception by deviating interaction attention from the original, without losing the perceptions on objects.

**Discussion on the Perturbation Bound.** We begin by assessing the efficacy of AIA across varying perturbation bounds, presenting the comparative results in Table 2. In typical adversarial attack scenarios, a higher $L_\infty$ constraint, such as transitioning from 10 to 20, often leads to greater success rates for the attacks. However, our observations reveal that the success rates of these attacks exhibit minimal fluctuations as the perturbation bound increases. Notably, the success rate for $L_\infty \leq 30$ is lower than that for $L_\infty \leq 10$ or $L_\infty \leq 20$. When targeting GEN-VLKT, the most effective attack is achieved with $L_\infty \leq 10$, followed by $L_\infty \leq 20$, while the least effective is with $L_\infty \leq 30$. On the other hand, for HOTR, the optimal attack performance is observed at $L_\infty \leq 20$, with lower success rates at both $L_\infty \leq 10$ and $L_\infty \leq 30$. This leads us to speculate that as the level of introduced noise, as represented by an increase in $L_\infty$, escalates, detrimental effects exert a stronger influence on the recognition of human bounding boxes, object bounding boxes, and object categories within the HOI task, subsequently compromising the efficacy of interaction-based attacks.

Table 2: Attack success rates with $L_\infty \leq \{10, 20, 30\}$.

| Iter | Target Model | Attack Method | $L_\infty \leq 10$ | $L_\infty \leq 20$ | $L_\infty \leq 30$ |
|---|---|---|---|---|---|
| 100 | GEN-VLKT | PGD(YOLOv3 & Faster R-CNN) | 67.65 | 68.47 | 64.04 |
| | | **Ours** | **76.35** | **70.94** | **69.95** |
| | HOTR | PGD(YOLOv3 & Faster R-CNN) | 56.33 | 62.03 | 60.13 |
| | | **Ours** | **67.41** | **71.52** | **69.30** |

To confirm this phenomenon, we've devised four Cases in Table 3, each governing changes in interaction, human bounding box, object bounding box, and object category. "Changed" signifies a necessary alteration, "IoU $\geq 0.5$" mandates an IoU score of at least 0.5, and "Disregarded" implies no specific attention is needed. In Case I, the conditions specified in Equation 6 are satisfied to determine a successful attack. In Case II, the rule for determining attack success is based on Case I but removes the requirement for the object category. In Case III and Case IV, the same logic applies to object bounding box and human bounding box, respectively. Through these experiments, we can analyze the factors that primarily influence the success of attacks on the target model. The bold numbers in Table 4 highlight how relaxing the indicator constraints affects the attack success rate when transitioning larger $L_\infty$. For example, when the number of adversarial iterations (Adv Iter) is set to 100 and the

Table 3: Successful attack verification cases based on four indicators.

|  | Interaction | Human Bounding Box | Object Bounding Box | Object category |
|---|---|---|---|---|
| Case I | 1 (Changed) | 1 (IoU $\geq$ 0.5) | 1 (IoU $\geq$ 0.5) | 1 (Unchanged) |
| Case II | 1 (Changed) | 1 (IoU $\geq$ 0.5) | 1 (IoU $\geq$ 0.5) | 0 (Disregarded) |
| Case III | 1 (Changed) | 1 (IoU $\geq$ 0.5) | 0 (Disregarded) | 0 (Disregarded) |
| Case IV | 1 (Changed) | 0 (Disregarded) | 0 (Disregarded) | 0 (Disregarded) |

Table 4: Attack success rate for four Cases on three iteration nodes, based on $L_\infty \leq \{10, 20, 30\}$. The bold data in the table indicate the maximum success rate corresponding to different $L_\infty$ constraints under the same Adv Iter and the same Case.

|  | Adv Iter = 1 | | | Adv Iter = 50 | | | Adv Iter = 100 | | |
|---|---|---|---|---|---|---|---|---|---|
| $L_\infty$ | 10 | 20 | 30 | 10 | 20 | 30 | 10 | 20 | 30 |
| Case I | **57.64** | 50.25 | 45.81 | **73.89** | 70.44 | 69.95 | **75.37** | 72.41 | 70.94 |
| Case II | **59.11** | 52.71 | 48.28 | **74.88** | 73.40 | 74.38 | 76.35 | **76.85** | 74.88 |
| Case III | **83.25** | **83.25** | 82.76 | 90.15 | **93.60** | 93.10 | 92.61 | **93.60** | 93.10 |
| Case IV | 87.19 | **89.16** | 88.67 | 91.63 | 95.07 | **95.57** | 93.60 | 95.07 | **95.57** |

indicator constraints are relaxed, the success rate significantly increases from 75.37% to 95.57% as $L_\infty$ increases from 10 to 30. This result confirms our previous speculation that noise plays a crucial role in influencing recognition, thereby compromising the efficacy of interaction-based attacks. More analysis is presented in appendix A table 6.

**Ablation Study**    To estimate the effectiveness of the proposed IAS and OCS modules in the AIA framework, we formulate the experimental setting as whether to use anchors, and whether to utilize object categories. As shown in Table 5, IAS improves the attack access rate with 1% by using anchors, and OCS further increases the rate up to 67.41%. We conclude that our proposed interaction attack pipeline successfully conducts the interaction attack task, and obtains more improvements with the modules for adversarial instance generation. More analysis is presented in appendix A table 7.

Table 5: Ablation study.

|  | Experiment set | | Result |
|---|---|---|---|
|  | Bounding box (IAS) | Category (OCS) | Iter = 100 |
| Adv RR | Random | Random | 65.19 |
| Adv RPD | Random | Prior-Distribution | 65.51 |
| Adv RPS | Random | Prior-Semantic | 66.46 |
| Adv AR | Anchor Area | Random | 66.14 |
| Adv APD | Anchor Area | Prior-Distribution | **67.41** |
| Adv APS | Anchor Area | Prior-Semantic | 65.82 |

## 5 CONCLUSION

Exploring the vulnerability of Deep Neural Networks (DNNs) against adversarial attacks has attracted significant attention and is extensively discussed in the field of artificial intelligence and its associated security. In contrast to adversarial attacks acting on image classification and object detection, this paper focuses on analyzing the fragility of interactions between contents in the scene. To address this issue, the paper introduces a novel black-box attacking framework named Adversarial Instance Attacks (AIA), which tampers interaction while preserving the perception of humans and objects. To the best of our knowledge, this work is the first time to discuss interaction attacks in a black-box manner. Extensive experiments show a large success rate in attacking interactions.

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

# A    APPENDIX

Table 6: Results from tests on Adv {RR, RPD, RPS, AR, APD, APS} according to the experimental sets in Table 3. Every four rows in the data correspond to a group of experiments. According to adversarial attack iteration, it is divided into three groups in 9 columns, representing 1, 50, 100 respectively. The bold parts in the table indicate the attack success rate corresponding to different $L_\infty$ constraints under the same Adv Iter and the same Case.

|  |  | Adv Iter = 1 | | | Adv Iter = 50 | | | Adv Iter = 100 | | |
|  |  | 10 | 20 | 30 | 10 | 20 | 30 | 10 | 20 | 30 |
|---|---|---|---|---|---|---|---|---|---|---|
| Adv RR | Case I | **55.17** | 52.71 | 47.78 | **76.35** | 69.95 | 69.46 | **76.35** | 71.92 | 70.44 |
|  | Case II | **56.16** | 54.68 | 49.26 | **76.35** | 72.41 | 75.37 | 76.85 | 74.88 | **76.35** |
|  | Case III | 83.74 | 83.74 | **84.73** | 92.61 | 92.61 | **93.10** | 93.10 | 93.10 | **94.09** |
|  | Case IV | 88.18 | 88.18 | **90.15** | 93.60 | 94.09 | **95.57** | 94.09 | 95.57 | **96.06** |
| Adv RPD | Case I | **61.88** | 50.25 | 50.25 | **75.74** | 69.95 | 69.46 | **76.24** | 72.41 | 71.92 |
|  | Case II | **62.87** | 51.72 | 52.22 | **77.23** | 73.40 | 71.92 | **77.72** | 75.37 | 74.88 |
|  | Case III | 83.17 | **84.24** | 82.76 | 91.58 | **93.60** | 93.10 | 92.08 | **94.09** | 93.60 |
|  | Case IV | 87.62 | 89.16 | **89.66** | 93.07 | **96.06** | 95.57 | 93.56 | **96.55** | 95.57 |
| Adv RPS | Case I | **56.65** | 52.71 | 47.78 | **73.89** | 70.44 | 70.44 | **74.88** | 71.92 | 72.91 |
|  | Case II | **57.64** | 55.17 | 49.26 | **74.38** | 71.92 | 73.89 | 76.35 | 74.38 | **76.85** |
|  | Case III | 83.74 | **85.22** | 82.76 | 91.63 | 92.61 | **93.10** | 93.10 | **94.09** | **94.09** |
|  | Case IV | 87.68 | **89.66** | 88.18 | 93.10 | 94.58 | **95.07** | 94.09 | **96.06** | 95.57 |
| Adv AR | Case I | **56.65** | 47.78 | 49.26 | **72.91** | 71.92 | 69.95 | **72.91** | **72.91** | 71.92 |
|  | Case II | **58.62** | 50.25 | 50.25 | **74.38** | 73.89 | 73.89 | 74.38 | 74.88 | **76.35** |
|  | Case III | **84.24** | 82.76 | 83.25 | 91.13 | **93.60** | 92.61 | 91.63 | **94.58** | 94.09 |
|  | Case IV | 87.68 | 88.67 | **89.16** | 92.61 | **96.06** | 94.58 | 93.10 | **97.04** | 96.06 |
| Adv APD | Case I | **57.64** | 50.25 | 45.81 | **73.89** | 70.44 | 69.95 | **75.37** | 72.41 | 70.94 |
|  | Case II | **59.11** | 52.71 | 48.28 | **74.88** | 73.40 | 74.38 | 76.35 | **76.85** | 74.88 |
|  | Case III | **83.25** | **83.25** | 82.76 | 90.15 | **93.60** | 93.10 | 92.61 | **93.60** | 93.10 |
|  | Case IV | 87.19 | **89.16** | 88.67 | 91.63 | 95.07 | **95.57** | 93.60 | 95.07 | **95.57** |
| Adv APS | Case I | **54.19** | 49.26 | 48.77 | **74.88** | 69.95 | 68.47 | **75.37** | 70.94 | 69.95 |
|  | Case II | **55.67** | 51.72 | 50.74 | **75.37** | 74.38 | 72.91 | **75.86** | **75.86** | 74.88 |
|  | Case III | 81.77 | 83.25 | **84.24** | 90.64 | 92.61 | **93.10** | 90.64 | 94.09 | **94.58** |
|  | Case IV | 86.70 | 88.67 | **91.13** | 92.61 | **95.07** | **95.07** | 92.61 | **96.06** | **96.06** |

Table 7: Ablation study results, with $L_\infty \leq 10$ as the constraint. Each row in the table represents the results of the adversarial attack iteration from 10 to 100 under the same Adv setting. The bolded parts are the items with the highest attack success rate among different Adv at the same adversarial attack iteration.

|  | 1 | 10 | 20 | 30 | 40 | 50 | 60 | 70 | 80 | 90 | 100 |
|---|---|---|---|---|---|---|---|---|---|---|---|
| Adv RR | 18.67 | 45.57 | 52.85 | 56.33 | 60.13 | 62.66 | 62.97 | 63.92 | 64.56 | 64.87 | 65.19 |
| Adv RPD | **19.62** | **46.20** | **54.43** | **57.28** | 59.18 | 60.44 | 62.66 | 63.92 | 64.24 | 64.24 | 65.51 |
| Adv RPS | 17.72 | 42.41 | 51.58 | 56.33 | 58.23 | 58.86 | 61.08 | 62.66 | 63.92 | 65.19 | 66.46 |
| Adv AR | 17.09 | 40.51 | 47.78 | 54.75 | **61.08** | **62.66** | **64.56** | **65.19** | 65.82 | 66.14 | 66.14 |
| Adv APD | 18.67 | 45.25 | 52.53 | 56.33 | 58.54 | 59.49 | 62.03 | 64.56 | 64.87 | **66.46** | **67.41** |
| Adv APS | 18.35 | 42.72 | 49.68 | 52.53 | 56.33 | 59.18 | 60.44 | 62.34 | 62.97 | 63.61 | 65.82 |

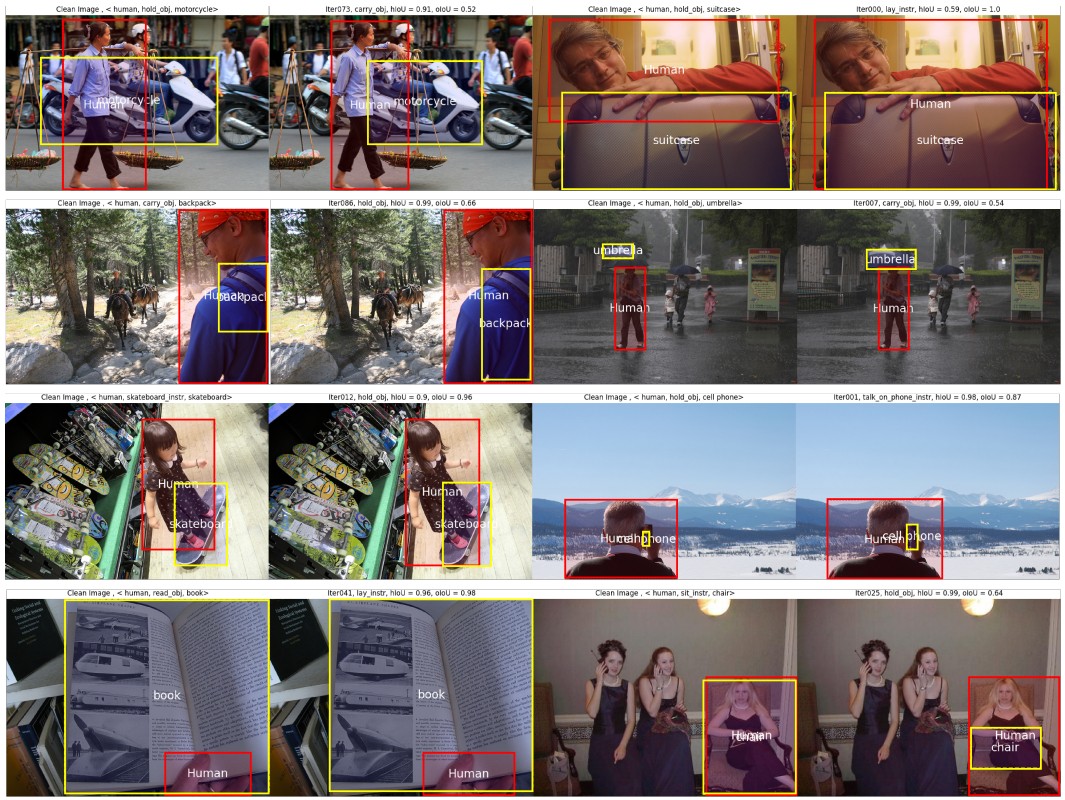

Figure 3: Here is the visualization of the attack results. Each group comprises two images: the clean image on the left and the adversarial image on the right. The title of each image includes the original prediction result, the IoU before and after the attack, and the interaction after the attack.

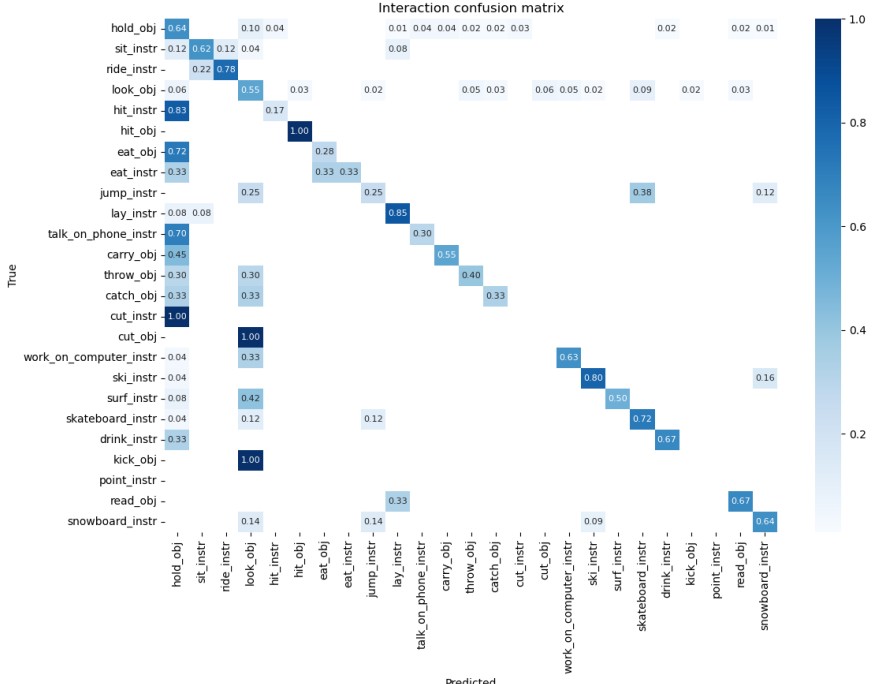

Figure 4: The confusion matrix analysis for interactions. The vertical axis represents the original interactions, while the horizontal axis represents the predicted interactions.

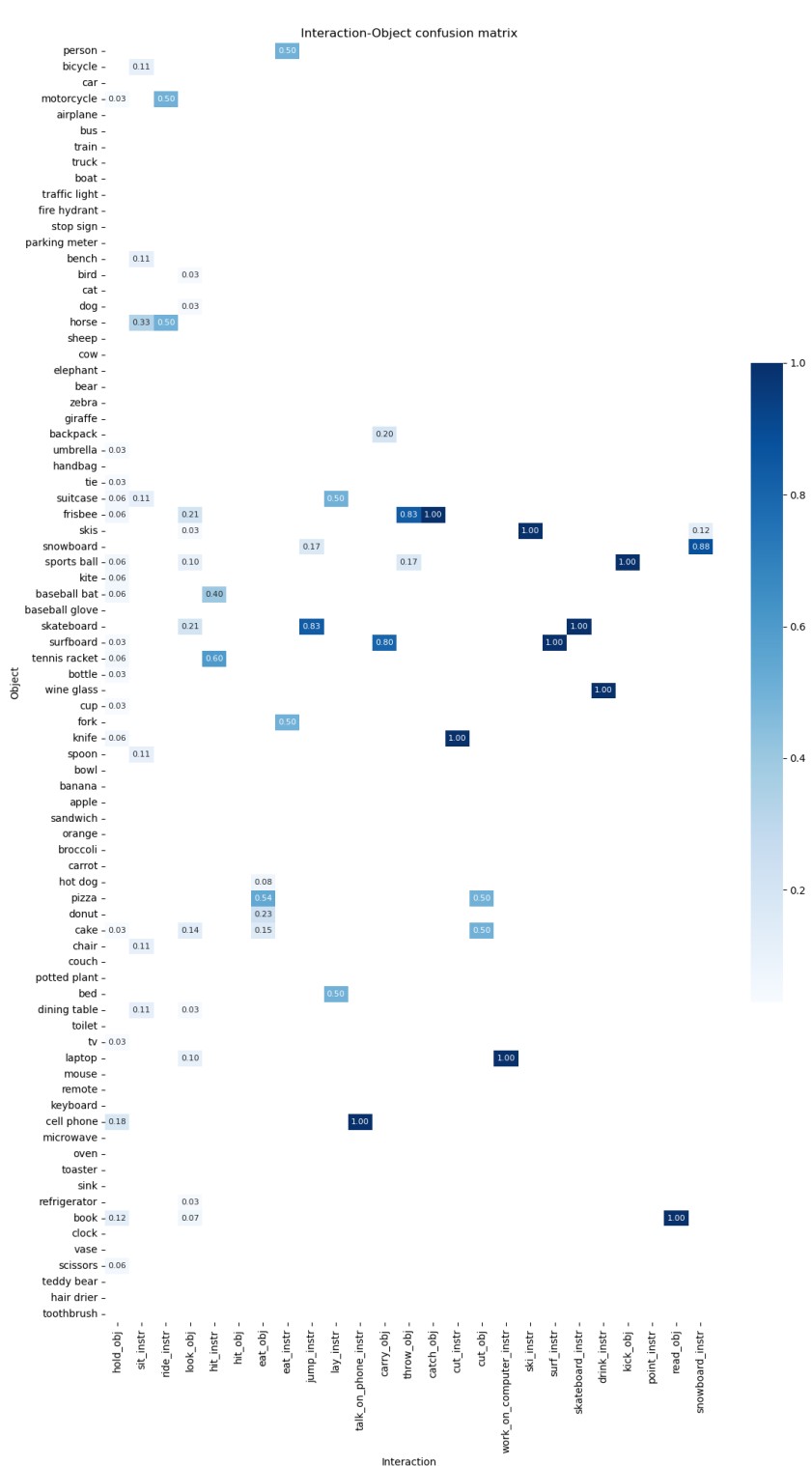

Figure 5: The frequency of occurrence on objects after misidentifying different interactions.

