# OpenReview forum: "Adversarial Instance Attacks for Interactions between Human and Object"
_ICLR.cc/2024/Conference — ICLR 2024 Conference Withdrawn Submission_

### Official Review · Reviewer_FvgA · 2023-10-27

**Soundness:** 3 good
**Presentation:** 2 fair
**Contribution:** 2 fair
**Rating:** 3
**Confidence:** 4

**Summary:**

The paper proposes a approach called Adversarial Instance Attacks (AIA) to attack the interaction predictions between humans and objects in the context of Human Object Interaction (HOI) tasks. The authors argue that existing adversarial attacks on object detection do not effectively extend to interactions, as interactions are more complex and require considering spatial and semantic relationships. AIA is designed to generate adversarial instances that distract the model's attention away from the original interaction, without disrupting the detection of human and object bounding boxes or their categories. The proposed AIA framework comprises three main modules: Interaction Area Sampling (IAS), Object Category Search (OCS), and Adversarial Instance Generation (AIG). The authors claim that AIA outperforms other attack methods in attacking interactions.

**Strengths:**

Innovative Approach: The paper introduces a novel approach to attacking interactions in HOI tasks, addressing a gap in the existing literature.

Experimental Results: The paper presents a thorough evaluation of the proposed AIA framework and compares it with other attack methods on various target and surrogate models. The results indicate that AIA is effective in attacking interactions.

Discussion on Perturbation Bounds: The paper provides an insightful analysis of the impact of different perturbation bounds on the success of attacks, shedding light on how noise affects interaction recognition.

**Weaknesses:**

Lack of Clarity: The paper is challenging to follow due to its complex technical language and insufficient contextual explanations. It assumes a high level of familiarity with the subject matter, making it less accessible to a broad audience.

Ambiguity in Terminology: Some terms and acronyms are not adequately defined or clarified, such as "Adversarial Instance Attacks," "Adversarial Interaction Attack," and "HOTR."

Missing Visual Aids: Given the complexity of the proposed framework, the paper would benefit from visual aids, diagrams, or flowcharts to help readers understand the different modules and their interactions.

Evaluation Metrics: The paper could benefit from a more detailed discussion of the evaluation metrics used and how they relate to the effectiveness of the attacks.

**Questions:**

1. Could you provide more clarity on the specific datasets used in the experiments and how they were collected or prepared?

2. Can you elaborate on the definition of "L" in the context of perturbation bounds? What does it represent, and how is it determined?

3. How does the AIA framework perform against state-of-the-art HOI models, and are there specific models it is more effective against?

4. What kind of perturbations or attacks are used in the experiments? Is there a specific method for generating these adversarial instances?

---

### Official Review · Reviewer_fSia · 2023-10-30

**Soundness:** 2 fair
**Presentation:** 1 poor
**Contribution:** 2 fair
**Rating:** 1
**Confidence:** 2

**Summary:**

The authors have introduced an adversarial attack problem that targets interactions between humans and objects. They evaluate this problem within the context of the Human Object Interaction (HOI) task, as proposed by Gkioxari et al. in 2018. Furthermore, the authors have successfully demonstrated an attacking framework for this novel problem

**Strengths:**

The paper proposes to tackle a new problem for adversarial attack for interactions between human and objects, based on the Human Object Interaction (HOI) task (Gkioxari et al., 2018), which is a new contribution to the field and might be interested to the adversarial attack community. However, the motivation is not clear when it is a difficult problem compared to existing attack problems. And the evaluation and writing quality are poor.

**Weaknesses:**

Deficient Writing Quality: The paper exhibits evident signs of hasty preparation, accompanied by numerous errors. To illustrate a few examples, some of which are highlighted in bold text:

* The reviewer could not understand “on the interaction between scene contents and **realize**’ based on”

* The math notations of {$ h_i^b, o_i^b, o_i^l, a_i $} are incorrect and confusing. It does not make sense to use a set { } for both coordinates and categories. And defining $a_i$ as the categories of the object is wrong.

* directly adding noise to the union area not only **drop the precise** of interaction perception but also **bring** ....

The motivation is unclear. The arguments within the sentences lack proper support. For instance, the sentence "a direct strategy to disturb the interaction is adding noises on the union area of human and object, following the feature extraction procedure in HOI. However, directly adding noise to the union area not only drop the precise of interaction perception but also bring interference in locating and classifying human and objects" lacks clarity and substantiation.

Understanding the paper's concept is challenging, primarily due to its poor writing quality. For example, the exact process for generating "N^q candidate interaction areas" is not explained and appears disconnected from the subsequent content. Upon reviewing Section 3.3, the reviewer found it to be incomplete, offering only a feature extraction process without any content related to sampling. The method for obtaining the extracted candidate predicate areas $A$ is also unclear.

The rationale behind distance calculation remains unclear, as does the definition of the midpoint function $f_{mid}()$.

"To address this issue, we employ random assignment to connect perplexing object categories to candidate predicate areas in order to generate the attack instances." However, the explanation for this random assignment is unclear.

In summary, while the paper may address an important problem of black-box adversarial attacks in scene interactions, its writing quality falls significantly below the standards of ICLR. Significant revisions are required. Therefore, the reviewer recommends clear rejection."


The equation (6) seems to contradict to the original claim of the paper  "our goal is to bring confusion on the
interaction of a pair of human and an object, without contaminating their detection results including
the human bounding box, object bounding box, and object category", as the object bounding boxes seem to be changed. Please provide more clarifications to it.

**Questions:**

Please refer to the above comments.

---

### Official Review · Reviewer_62sG · 2023-10-31

**Soundness:** 3 good
**Presentation:** 2 fair
**Contribution:** 2 fair
**Rating:** 3
**Confidence:** 4

**Summary:**

In this paper, the authors propose a black-box attack method against Human Object Interaction (HOI). The proposed method of Adversarial Instance Attacks (AIA) consists of an Interaction Area Sampling (IAS) module for finding the vulnerable areas, an Object Category Search (OCS) module for assigning the confusing category, and an Adversarial Instance Generation (AIG) module for attacking effects. The experiment validations on the V-COCO datasets are given to demonstrate its effectiveness on various models.

**Strengths:**

1. The proposed attack method is conducted in a new area, Human Object Interaction.
2. Choosing the fragility of one image to attack is an interesting and reasonable idea for Human Object Interaction.

**Weaknesses:**

1. For the experiments, the comparisons between I-FGSM and PGD on detectors and the proposed method are unfair. The authors use the transfer results by I-FGSM and PGD to attack HOI, where the adversarial examples based on the architecture of detectors (e.g., Faster RCNN and YOLO ) are generated. Thus, these methods have a weak transferability across the different tasks. However, the proposed method has more knowledge about the attacked task instead of only detectors.
2. The tasks of HOI highly rely on the results of detections. The existing attacks for detection can be directly employed in this task, leading to awful results. From the proposed method, the attacks aim to fool the interactions between the human and object, instead of the human and the object itself. If so, wrong detection results and boxes easily lead to wrong interactions. Thus, those attacks against object detection, especially for fooling the detection boxes should be involved in comparisons. Besides, why only attacking the interaction in HOI is worthy of investigating?
3. Expect for the illustration in Figure 1, more visualizations with various methods (e.g., PGD and I-FGSM) should be involved to demonstrate its imperceptibility and effectiveness.
4. There are some minor problems.
- In the caption of Figure 1, ‘remaining0’ seems a typo.
- Some formats of references are wrong and inconsistent.

**Questions:**

1. The comparison between I-FGSM and PGD on detectors and the proposed method for HOI is unfair as the prior knowledge is different.
2. Discuss the motivation for only attacking the interaction in HOI, and give some attack results on only attacking the detectors.
3. Give more visual comparisons between various methods.

---

### Official Review · Reviewer_L4HX · 2023-11-02

**Soundness:** 3 good
**Presentation:** 3 good
**Contribution:** 2 fair
**Rating:** 5
**Confidence:** 3

**Summary:**

This paper proposed a new concept for adversarial attack on Human Object Interaction (HOI). Different from adversarial attack on object classification or detection, there was less work focused on HOI attack. In this paper, the main idea is to find a proper region in an image, where an invisible object is inserted and which will mislead the HOI classifier. To this end, three modules are proposed, i.e., Interaction Area Sampling module, Object Category Search module, and Adversarial Instance Generation module.

**Strengths:**

+ A novel adversarial attack is defined for human object interaction recognition task. HOI recognition task is a high-level task with respect to object classification or detection tasks. It motivation of this paper makes sense which is to insert an inviable object to mislead the HOI classifier.

**Weaknesses:**

- My main concern is that the proposed attack is somewhat an untargeted attack for an HOI recognition system, where we cannot choose the target HOI category, instead, the target HOI category is determined by the method via finding the most confusing interaction. I am not sure the whether such attack will happen in real applications or not.
- The proposed method is straightforward, which is combination of existing techniques such as Targeted adversarial Objectness Gradient attacks (TOG) in Adversarial Instance Generation module.

**Questions:**

Can we conduct a targeted adversarial attack for HOI recognition, which is that we can explicitly choose the target HOI category we want the attack achieves.

---

### Meta-Review · Area_Chair_xxoc · 2023-12-05

**Metareview:**

Four experts reviewed the paper, and none was supportive. There was no rebuttal.

**Justification For Why Not Higher Score:**

No reviewer was supportive

**Justification For Why Not Lower Score:**

No lower score available

---

### Decision · Program_Chairs · 2024-01-16

Reject